# Genome-Wide Identification of the *PEBP* Family Gene in Three *Cymbidium* Species and Its Expression Patterns Analysis in *C. ensifolium*

**Jinliao Chen** †, **Fei Wang** †, **Yangting Zhang, Ruiyue Zheng, Xiaopei Wu, Ye Ai, Sagheer Ahmad,**
**Zhongjian Liu * and Donghui Peng ***

Key Laboratory of National Forestry and Grassland Administration for Orchid Conservation and Utilization at Landscape Architecture and Art, Fujian Agriculture and Forestry University, Fuzhou 350002, China; fjchenjl@126.com (J.C.); 19559119005@163.com (Y.Z.); wuxp7872@163.com (X.W.); sagheerhortii@gmail.com (S.A.)
* Correspondence: zjliu@fafu.edu.cn (Z.L.); fjpdh@fafu.edu.cn (D.P.)
† These authors contributed equally to this work.

**Abstract:** The *PEBP* gene family is involved in many biological processes in plants, including plant growth and development, flowering regulation, light response, and abiotic stress response. But there is little information about the role of the *PEBP* gene family in *Cymbidium* species. In this study, we identified 11, 9, and 7 *PEBP* genes in *C. ensifolium*, *C. sinense*, and *C. goeringii*, respectively, and mapped them to the chromosomes. We also studied the physicochemical characteristics of the proteins encoded by these *PEBP*s and analyzed their intra-species collinearity, gene structure, conserved motifs, and cis-acting elements. Furthermore, a total of forty PEBP genes from *C. sinense*, *C. ensifolium*, *C. goeringii*, *Phalaenopsis*, and *Arabidopsis* were divided into three clades based on the phylogenetic tree. The expression patterns of 11 *PEBP* genes in different tissues and organs of *C. ensifolium* were analyzed based on transcriptome data, indicating that the *CePEBP*s might play an important role in the growth and development, especially in the flower bud organs (1–5 mm). *CePEBP5* plays an indispensable role in both the vegetative and reproductive growth cycles of *C. ensifolium*. *CePEBP1* is essential for root development, while *CePEBP1*, *CePEBP3*, *CePEBP5*, and *CePEBP10* regulate the growth and development of different floral organ tissues at various stages. The findings of this study can do a great deal to understand the roles of the *PEBP* gene family in *Cymbidium*.

**Keywords:** *PEBP* gene family; gene family analysis; expression pattern; *Cymbidium* species

## 1. Introduction

Phosphatidylethanolamine-binding proteins (PEBPs) contain a conserved PEBP structural domain and exhibit a strong affinity for phosphatidylethanolamine [1]. They play an important role in regulating flowering, seed development, and germination in plants [1–4]. The PEBP gene family can be divided into three clades in angiosperms: MOTHER OF FT AND TFL1-like (MFT-like), FLOWERING LOCUS T-like (FT-like), and TERMINAL FLOWERING 1-like (TFL1-like) [5]. FT-like and TFL1-like genes are reported only in gymnosperms and angiosperms, whereas MFT-like genes can be traced back to the origin of land plants. Therefore, MFT-like genes are the common ancestor of FT-like and TFL1-like genes [4–6].

Despite the high degree of sequence similarity between members of the PEBP gene family, their functions are not identical. MFT-like genes play an important role in the regulation of the development, germination, and dormancy of seeds [7–9]. It has been demonstrated that the regulation of flowering time and morphogenesis can be controlled by most members of the FT-like gene and TFL1-like genes [10]. The proteins encoded by FT-like genes act as flowering promoters in plants [8,11,12]. In *Arabidopsis thaliana*, FT acts as a floral signal transducer, moving from leaves, passing through the phloem to the shoot

apical meristem, and binding to Floring Locus D (FD) proteins. It promotes the expression of downstream flowering-related genes (such as AP1), thereby regulating the flowering process in plants. The TFL1 subfamily consists of TFL1, CENTRORADIALIS (CEN), and BROTHER OF FT AND TFL1 (BFT). TFL1-like genes have high sequence similarity to FT-like genes, but they have opposite functions. They inhibit flowering by binding to the bZIP-type transcription factor FD and maintain the infinite growth of inflorescence meristematic tissue [13–17]. There is 60% homology in the amino acid sequences between FT and TFL1, but only a few amino acids need to be changed to convert FT from a floral promoter to TFL1, a floral repressor. This is mainly due to inconsistencies in two key amino acid sites, Tyr85 in FT and His88 in TFL1 [18,19]. In addition, the 14 amino acid fragment LGRQTVYAPGWRQN and the triplet LYN in exon 4 of FT/TFL1-like also play important roles in the opposite function of FT/TFL1-like [20].

*C. ensifolium*, *C. goeringii*, and *C. sinense* are the most significant ornamental orchids because of their beautiful and unique flowers. They have a long history of cultivation and are loved by consumers in China [21]. The *PEBP* gene family has not yet been systematically analyzed by bioinformatics, although some members have been identified and studied in *Cymbidium*. Given the considerable role of *PEBP* genes in regulating plant flowering, seed development, and germination, this study utilized bioinformatics methods to perform genome-wide identification of three *Cymbidium* species in Orchidaceae. In this study, we identified twenty-seven members of the PEBP gene family in three *Cymbidium* species, determined their chromosomal localization, constructed phylogenetic trees, and analyzed the gene structure, conserved motifs, and cis-acting element types. Additionally, we analyzed 11 members of the *CePEBP* genes in different tissues of *C. ensifolium*. Our findings further elucidate the functions of *PEBP* genes in the flowering and vegetative development of three *Cymbidium* species and provide suggestions for improvement and the creation of new varieties.

## 2. Materials and Methods

### 2.1. Data Sources

The genome sequences and annotation information of *C. ensifolium*, *C. goeringii*, and *C. sinense* were retrieved from their whole-genome sequencing data [22–24]. The protein sequence of PEBP gene family of *A. thaliana* was retrieved from the Arabidopsis Information Resource (TAIR, https://www.arabidopsis.org/, accessed on 20 August 2023). The *Phalaenopsis* 'Little Gem Stripes' data (PhFT1 (Peq009747), PhFT2 (Peq006920), PhFT3 (Peq017805), PhFT4 (Peq012163), PhFT5 (Peq006349), PhFT6 (Peq009750), and PhMFT (Peq004653)) were downloaded from National Center for Biotechnology Information (NCBI, https://www.ncbi.nlm.nih.gov/, accessed on 20 August 2023) [25].

### 2.2. Identification and Physicochemical Properties of PEBP Genes from Three Cymbidium Species

The conserved domain of PEBP (PF01161) was downloaded from Pfam. Using the BLAST and Simple HMM Search functions of TBtools (version 1.132). The PEBP family members of three *Cymbidium* species were identified from the genome databases [26,27]. The screening parameter had an E-value lower than $1 \times 10^{-5}$. Then, NCBI CD-Search (https://www.ncbi.nlm.nih.gov/Structure/cdd/wrpsb.cgi, accessed on 20 August 2023) [28–30] and SMART (http://smart.embl.de/, accessed on 20 August 2023) [31] websites were used to analyze the structure of candidate PEBP proteins and eliminate incomplete and redundant protein sequences. The PEBP genes were named and classified according to the naming rules of *A. thaliana*. Finally, the ExPASy website (https://www.expasy.org/, accessed on 20 August 2023) was used to calculate the amino acid (aa), isoelectric point (pI), molecular weight (MW), grand average hydrophilicity (GRAVY), instability index (II), and lipid index (AI) of the PEBP proteins [32,33].

### 2.3. Phylogenetic Analysis of PEBP Genes

The protein sequences of 7 *PEBPs* from *C. goeringii*, 11 *PEBPs* from *C. ensifolium*, 9 *PEBPs* from *C. sinense*, 6 *PEBPs* from *A. thaliana*, and 7 *PEBPs* from *Phalaenopsis* 'Little Gem Stripes' were

imported in MEGA 7.0 software [34]. A total of 40 protein sequences were aligned using the MUSCLE program with default parameters. The phylogenetic tree of *PEBPs* was constructed based on maximum likelihood (ML), with Bootstrap parameters set to 1000 and partial deletion to 75% [34]. For better visualization, the phylogenetic tree was processed using the online software iTOL 6.8.2 (https://itol.embl.de/itol.cgi, accessed on 20 August 2023) [35].

*2.4. Chromosome Distribution and Collinear Correlation of PEBP Genes in Three Cymbidium Species*

The visualization and analysis of chromosomal localization of *PEBP* genes in three *Cymbidium* were conducted using Tbtools software, utilizing the genome and annotation files of *C. ensifolium*, *C. goeringii*, and *C. sinense*. In addition, the genomic data of the three *Cymbidium* species were analyzed in collinearity analysis using the One-Step MCScanx program in Tbtools [27]. In Tbtools, the replication patterns of three *Cymbidium* species were visualized using Advance Circos [27].

*2.5. Gene Structure and Conserved Motif Analysis of PEBP Gene*

The conserved domains of the *PEBP* genes of three *Cymbidium* species were predicted using the CDD website (https://www.ncbi.nlm.nih.gov/cdd, accessed on 20 August 2023) [29,30], and the motifs of these *PEBP* genes were analyzed using the MEME website (https://meme-suite.org/meme/tools/meme, accessed on 20 August 2023) [36]. Ten motifs were followed, and the other was the default value.

*2.6. Sequence Analysis of PEBP Gene Promoter*

The sequence of 2000 bp upstream of the transcription start site was extracted using Tbtools as the promoter sequence of the *PEBP* genes in three *Cymbidium* species [26,27]. Additionally, potential cis-acting elements on the promoter sequences were predicted using PlantCARE website (https://bioinformatics.psb.ugent.be/webtools/plantcare/html/, accessed on 20 August 2023) [37]. Then, the prediction results were classified and analyzed using Excel 2019 and Tbtools software.

*2.7. Expression Pattern and qRT-PCR Analysis*

The expression patterns of 11 *CePEBP* genes were analyzed to investigate the potential impact of *PEBP* genes across different organs of *C. ensifolium*. The sampled organs for investigation included the root, leaf, buds of various sizes (1–5 mm, 6–10 mm, and 11–15 mm), petal, lip, sepal, pedicel, and gynostemium. Three biological replicates were analyzed, each of which was a pooled sample from five plants. We conducted transcriptome analysis on all ten samples, calculating the fragments per kilobase of transcript per million mapped reads (FPKM) [22]. The heatmap showed the patterns of expression using TBtools (version 1.132) [27].

Quantitative real-time PCR (qRT-PCR) was used to further analyze the expression patterns of the *CePEBPs*. The root, leaf, buds of various sizes (1–5 mm, 6–10 mm, and 11–15 mm), petals, lip, sepal, pedicel, and gynostemium at blooming period were sampled from *C. ensifolium* 'Longyansu' planted at the Fujian Agriculture and Forestry University. Primer Premier 5.0 software was used to design primers. The details of the primers and reference genes are listed in Supplementary Table S3. Total RNA was extracted by the TIANGEN DP441 Reagent Kit (Tiangen, Beijing, China). A HiScript III 1st Strand cDNA Synthesis Kit (+gDNA wiper; Vazyme, Nanjing, China) was used to reverse-transcribe RNA to cDNA. Based on the Taq Pro Universal SYBR qPCR Master Mix kit (Vazyme, Nanjing, China), the ABI 7500 Real-Time System (Applied Biosystems, Foster City, CA, USA) was used to analyze the RT-qPCR. Finally, the $2^{-\Delta\Delta CT}$ method was used to calculate the expression level [26,33,38].

**3. Results**

*3.1. Identification and Sequence Analysis of PEBP Genes in Three Cymbidium Species*

The basic information and physicochemical properties of the *PEBP* genes for three *Cymbidium* species are shown in Table 1. A total of 11, 9, and 7 *PEBP* genes were

found in *C. ensifolium*, *C. sinense*, and *C. goeringii*, respectively. Based on the sequential distribution on chromosomes, these 27 *PEBP* genes were named *CePEBP1-11*, *CsPEBP1-9*, and *CgPEBP1-7*, respectively. A sequence analysis of the encoded proteins showed that the physicochemical properties of amino acids, isoelectric point, molecular weight, grand average of hydropathicity, aliphatic index, and instability index of the *PEBP* genes in three *Cymbidium* species differed significantly (Table 1). The deduced protein length (AA) of *PEBP* genes ranged from 66 (*CgPEBP3*) to 379 (*CsPEBP9*) amino acids. The isoelectric point (pI) values of the 27 *PEBP* genes in *Cymbidium* ranged from 5.13 (*CgPEBP1*) to 10.75 (*CgPEBP3*). Among them, 6 PEBP proteins had an acidic pI below seven, while the 21 PEBP proteins with a pI higher than seven were alkaline. The grand average of hydropathicity (GRAVY) values of *PEBP* genes were ranged from −0.577 (*CgPEBP3*) to −0.118 (*CePEBP2*), with all GRAVY values of less than 0, indicating that they were hydrophilic. The molecular weight (Mw) ranged from 7806.99 kD (*CgPEBP3*) to 42330.61 kD (*CsPEBP9*), with the aliphatic index (AI) between 66.52 (*CgPEBP3*) and 89.15 (*CePEBP4*). The maximum instability index (II) value was 58.26 (*CePEBP2*), and the minimum value was 30.19 (*CgPEBP4*).

**Table 1.** PEBP gene family protein properties table from three *Cymbidium* species.

| Gene Name | Gene ID | Protein Length (AA) | Isoelectric Point (pI) | Molecular Weight (Mw) | Grand Average of Hydropathicity (GRAVY) | Aliphatic Index (AI) | Instability Index (II) |
|---|---|---|---|---|---|---|---|
| *CePEBP1* | JL006795 | 176 | 6.42 | 19,848.39 | −0.311 | 80.74 | 43.37 |
| *CePEBP2* | JL020923 | 173 | 7.74 | 19,256.21 | −0.118 | 83.29 | 58.26 |
| *CePEBP3* | JL010014 | 173 | 9.06 | 19,607.53 | −0.202 | 82.14 | 45.42 |
| *CePEBP4* | JL026838 | 189 | 6.73 | 21,260.35 | −0.151 | 89.15 | 46.08 |
| *CePEBP5* | JL020421 | 174 | 6.42 | 19,926.58 | −0.355 | 75.52 | 35.95 |
| *CePEBP6* | JL027939 | 101 | 5.62 | 10,995.31 | −0.265 | 73.27 | 41.88 |
| *CePEBP7* | JL002228 | 174 | 9.03 | 19,523.09 | −0.375 | 77.82 | 48.29 |
| *CePEBP8* | JL001165 | 177 | 9.18 | 20,180.03 | −0.227 | 83.11 | 45.55 |
| *CePEBP9* | JL013430 | 178 | 8.48 | 20,082.78 | −0.39 | 74.44 | 42.64 |
| *CePEBP10* | JL027407 | 112 | 5.27 | 12,308.11 | −0.14 | 86.88 | 52.12 |
| *CePEBP11* | JL028740 | 183 | 6.12 | 20,580.41 | −0.248 | 81.97 | 48.35 |
| *CgPEBP1* | GL13937 | 236 | 5.13 | 26,017.24 | −0.323 | 80.04 | 48.92 |
| *CgPEBP2* | GL01335 | 176 | 6.42 | 19,848.39 | −0.311 | 80.74 | 43.37 |
| *CgPEBP3* | GL28974 | 66 | 10.75 | 7806.99 | −0.577 | 66.52 | 54.87 |
| *CgPEBP4* | GL14129 | 125 | 6.83 | 14,228.19 | −0.331 | 82.56 | 30.19 |
| *CgPEBP5* | GL07645 | 174 | 9.03 | 19,522.14 | −0.333 | 80.06 | 48.67 |
| *CgPEBP6* | GL00658 | 201 | 7.8 | 22,704.79 | −0.341 | 77.06 | 38.75 |
| *CgPEBP7* | GL09595 | 181 | 6.73 | 20,394.24 | −0.229 | 82.87 | 49.19 |
| *CsPEBP1* | cymsin_Mol026710 | 118 | 5.34 | 13,110.05 | −0.251 | 85.85 | 51.29 |
| *CsPEBP2* | cymsin_Mol020839 | 189 | 6.42 | 21,659.53 | −0.355 | 79.31 | 37.43 |
| *CsPEBP3* | cymsin_Mol012759 | 173 | 9.06 | 19,598.52 | −0.203 | 82.14 | 45.59 |
| *CsPEBP4* | cymsin_Mol020552 | 187 | 7 | 20,394.34 | −0.184 | 73.48 | 57.35 |
| *CsPEBP5* | cymsin_Mol006878 | 190 | 6.08 | 21,404.16 | −0.297 | 80.95 | 41.78 |
| *CsPEBP6* | cymsin_Mol006013 | 243 | 6.08 | 27,504.27 | −0.363 | 80.16 | 42.56 |
| *CsPEBP7* | cymsin_Mol018868 | 174 | 9.03 | 19,523.09 | −0.375 | 77.82 | 48.29 |
| *CsPEBP8* | cymsin_Mol003216 | 178 | 9.18 | 20,292.16 | −0.23 | 83.2 | 42.1 |
| *CsPEBP9* | cymsin_Mol017371 | 379 | 9.57 | 42,330.61 | −0.181 | 84.62 | 52.69 |

Note: Ce: *Cymbidium ensifolium*; Cg: *C. goeringii*; Cs: *C. sinense*.

### 3.2. Chromosome Localization and Collinearity Analysis of PEBP Genes in Three Cymbidium Species

The relationship between the location of the *PEBP* genes on the chromosomes and the collinearity within three *Cymbidium* species is shown in Figure 1. The *PEBPs* are exclusively located on the partial chromosomes of three *Cymbidium* species and are dispersed. Ten *CePEBPs* were unevenly distributed on seven chromosomes (Chr02, 05, 06, 09, 11, 14, and 17). Chromosome 02,05,09 contained two genes, while each of the remaining chromosomes contained one gene. *CePEBP9* was localized to the unanchored scaffold, named Scaffold5149. Chromosome localization analysis in *C. sinense* showed that nine *CsPEBPs* were unevenly distributed on seven chromosomes (Chr02, 06, 07, 08, 11, 13, and 17). The chromosome 08 contained three genes (*CsPEBP4*, *CsPEBP5*, and *CsPEBP6*), while the other chromosomes contained one gene each. Six *CgPEBPs* were unevenly distributed on four chromosomes (Chr01, 05, 11, and 16). Chromosome 01 contained three genes (*CgPEBP1*, *CgPEBP2*, and *CgPEBP3*), whereas the other chromosomes contained one gene each. *CgPEBP7* was localized to the unanchored scaffold, named Scaffold10 (Figure 1A).

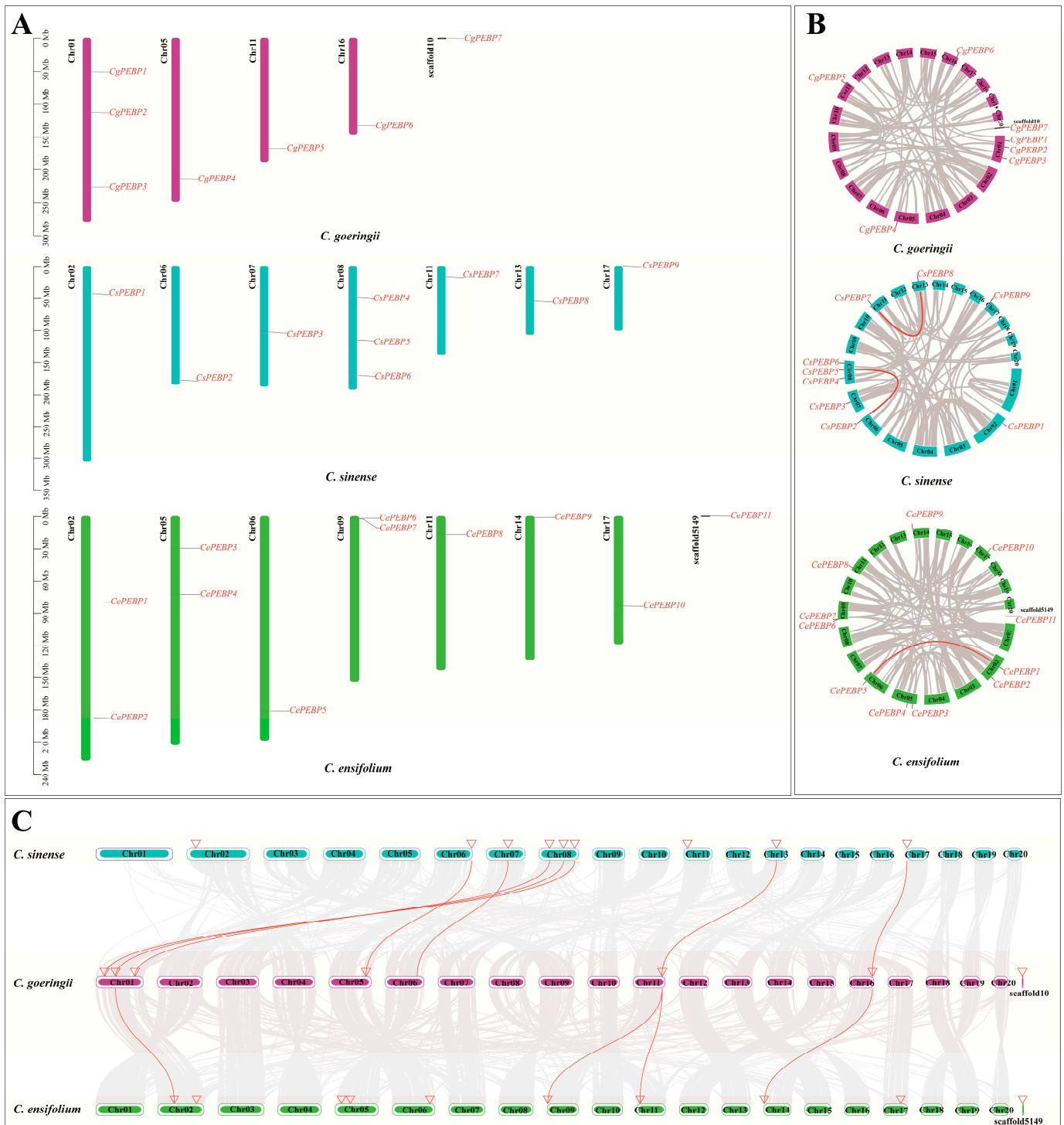

**Figure 1.** (**A**) The position of PEBP gene family members on chromosomes in three *Cymbidium* species. The left-hand scale is used to estimate the length of chromosomes. (**B**) Intraspecific collinearity of the *PEBP* genes in the three *Cymbidium* species. (**C**) Interspecific collinearity relationship between PEBP gene family members and *C. goeringii*, *C. ensifolium*, and *C. sinense*. The chromosomes of *C. goeringii*, *C. ensifolium*, and *C. sinense* are marked with different colors. Red lines connect the collinear relationship between PEBP gene family members of different species, and the location of *PEBPs* is represented by the red triangle. Circles of different colors represent different *Cymbidium* species.

The analysis of collinearity between genes revealed replication relationships between them. Within *C. ensifolium*, the *PEBP* genes had one collinear relationship, which was

between *CePEBP5* on Chr06 and *CePEBP1* on Chr02 (Figure 1B). There were two collinear relationships in *C. sinense*, which were *CsPEBP2* on Chr06 and *CsPEBP5* on Chr08, and *CsPEBP7* on Chr011 and *CsPEBP8* on Chr11, respectively. They exhibited similar conserved motifs and gene arrangements. No collinear relationship was detected in *C. goeringii* (Figure 1B). Figure 1C shows the collinear relationships of *PEBP* genes in three *Cymbidium* species. The analysis results indicated that *C. goeringii* shared seven collinearities with *C. sinense* and four collinearities with *C. ensifolium*. Among the three species, *CgPEBP* was most closely related to *CsPEBP*.

### 3.3. Phylogenetic Relationship Analysis of PEBP Genes

To analyze the phylogenetic relationships of *PEBPs* and other homologous genes in three *Cymbidium* species, we constructed a maximum likelihood (ML) phylogenetic tree using the amino acid sequences of 6 *AtPEBP* proteins, 7 *PhPEBP* proteins, 7 *CgPEBP* proteins, 9 *CsPEBP* proteins, and 11 *CePEBP* proteins (Figure 2). The phylogenetic trees showed that the forty PEBP proteins were categorized into three subfamilies: FT, TFL1, and MFT. Among these, twenty-two *PEBP* genes belonged to the FT subfamily for three *Cymbidium* species (six *CgPEBP*, seven *CsPEBP*, and nine *CePEBP*, respectively), and the number of members of the TFL1 subfamily was the lowest (two members, *CePEBP3* and *CsPEBP3*, respectively). Moreover, the TFL1 gene was absent in *C. goeringii*. The *PEBPs* of the three *Cymbidium* species were classified into four types (I-IV) based on the topology of the phylogenetic tree. The like-I clade consists of genes that are very similar to the FT-like genes in *A. thaliana*.

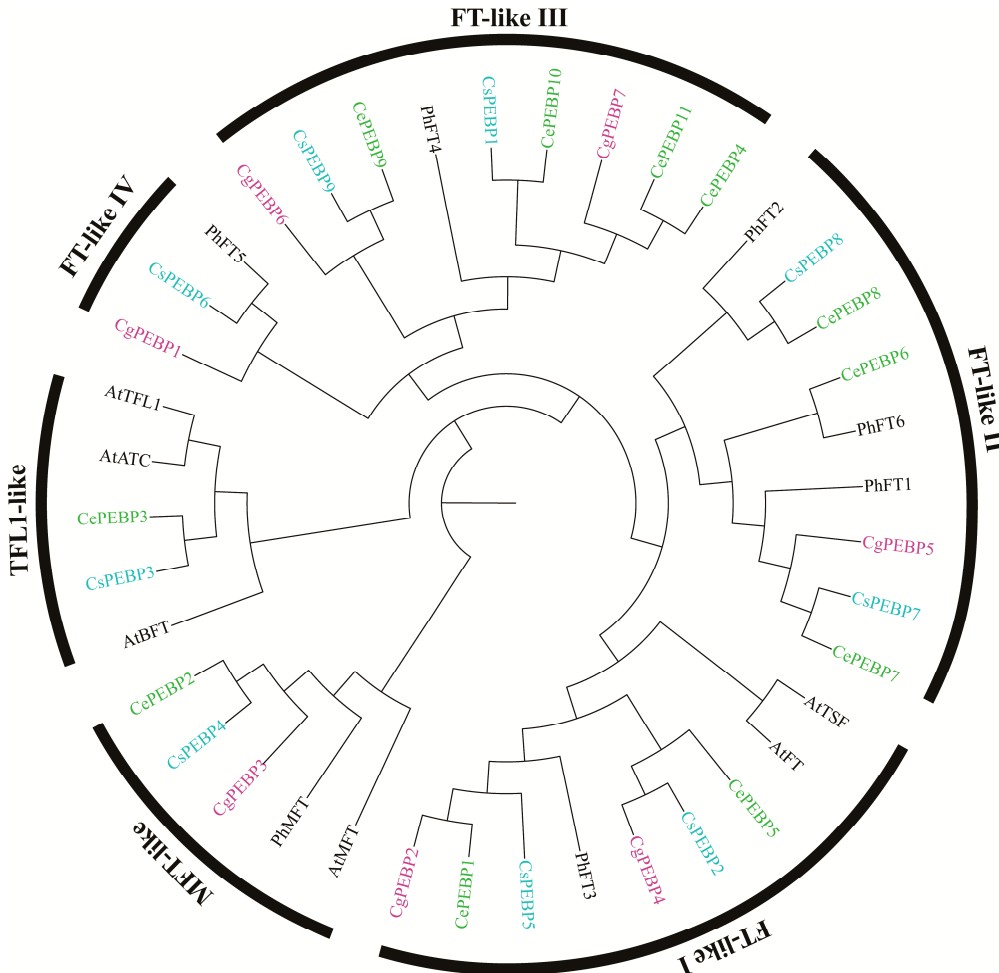

**Figure 2.** Phylogenetic tree of the 40 PEBP proteins from *Cymbidium goeringii*, *C. ensifolium*, *C. sinense*, *A. thaliana*, and *Phalaenopsis hybrid*. Circles of different colors represent different *Cymbidium* species.

### 3.4. Conserved Motif and Gene Structure Analysis of PEBP Gene Family

To understand the gene structure of *PEBP*s in three *Cymbidium* species, we predicted ten conserved motifs of *PEBP* genes by MEME and demonstrated the exon intron structure using Tbtools. The result showed that the majority of *PEBP* genes exhibited six conserved motifs, numbered one to six. Members of the same subfamily within the PEBP family share similar conserved motifs. Additionally, motifs 7 and 9 were exclusively observed in MFT-like genes. Except for *CePEBP10* and *CsPEBP1*, almost all *PEBP* genes contained at least four conserved motifs (Figure 3A). Motif 1 (e-value = 2.0e − 723) contained a conserved motif D-P-D-X-P and its critical AA84 amino acid residue (Y). Motif 2 (e-value = 3.6e − 708) contained the critical AA139 amino acid residue (Q). Motif 4 (e-value = 8.3e − 285) contained a conserved motif G-X-H-R, and motif G-X-H-R had a strong effect on the Ile (I) residue with a preference (Figure 3B, 3C). The gene structure analysis indicated that 17 *PEBP*s (accounting for 63%) contained four exons and three introns, and four *PEBP*s (accounting for 15%) contained three exons and two introns. Three *PEBP*s (accounting for 11%) contained five exons and four introns, and three *PEBP*s (accounting for 11%) contained six exons and five introns. All *PEBP* genes had between one and five introns, with *CsPEBP9* having the longest intron and *CgPEBP3* having only one intron (Figure 3B).

The PEBP gene family had two key amino acid (AA) residues at the AA85 (Tyr, Y) and AA140 (Gln, Q) positions in *Arabidopsis* [18,20]. We performed the protein alignment of *PEBP* homologs from three *Cymbidium* species (Figure 3D). In these three *Cymbidium* species, Tyr (Y) at AA85 was replaced by Cys (C) and His (H) in MFT-like (*CgPEBP3*, *CsPEBP4*, and *CePEBP2*) and TFL1-like (*CsPEBP3*, *CePEBP3*), respectively. In the FT subfamily, Tyr (Y) at AA85 was replaced by His (H) and Leu (L) in three genes (*CgPEBP6*, *CsPEBP9*, *CePEBP9*) and five genes (*CgPEBP7*, *CePEBP10*, *CePEBP11*, *CePEBP4*, *CsPEBP1*) of FT-like III, respectively. The key amino acid residues of other PEBP genes in the FT subfamily were highly conserved at AA85. In addition, another key amino acid residue at the AA140 (Gln, Q) positions of *PEBP* genes was replaced by Asp (D) in the TFL1-like subfamily and replaced by Glu (E) and His (H) in the FT-like II tapy, respectively, and in *CgPEBP4* by Lys (K). Key amino acid residues of other *PEBP* genes were highly conserved at AA140. The amino acid comparison showed that the functions of these *PEBP* genes might be largely conserved (Figure 3D).

### 3.5. Cis-Element Analysis of Three Cymbidium Species

We extracted the 2000 bp sequence upstream region of each gene in the *PEBP* genes for three *Cymbidium* species and predicted cis-acting elements using the PlantCARE databases. There were 690 predicted cis-acting elements in three *Cymbidium* species, and *C. ensifolium* had the most cis-elements (273/690), followed by *C. sinense* (239/690) and *C. goeringii* (178/690). These were categorized into six groups: light-responsive elements (317), hormone-responsive elements (213), developmental-associated elements (43), environmental stress-related elements (88), site-binding-associated elements (22), and promoter-associated elements (7). Among them, the maximum number of light-responsive elements were Box 4 (94/317, 29.65%) and G-box (62/317, 19.55%), followed by TCT-motifs (24/317) and GT1-motifs (223/317). Among the phytohormone response elements, a higher number of ABRE (58/213, 27.23%), CGTCA-motif (39/213, 7.8%), and TGACG-motifs (39/213, 7.8%), were associated with abscisic acid response and MeJA, respectively. The remaining elements were associated with the salicylic acid response. Among the plant growth- and development-related response elements, GCN4-motif (14/43, 32.55%) and O2-site (12/43, 27.91%) were associated with circadian rhythm control and arginine metabolism, respectively. In contrast, other elements such as CAT-box motifs, circadian, and RY-elements were associated with phloem tissue expression and seed development. Thus, the *PEBP* genes of the three orchids were mainly associated with light response and the regulation of tissue metabolism (Figure 4, Supplementary Table S1).

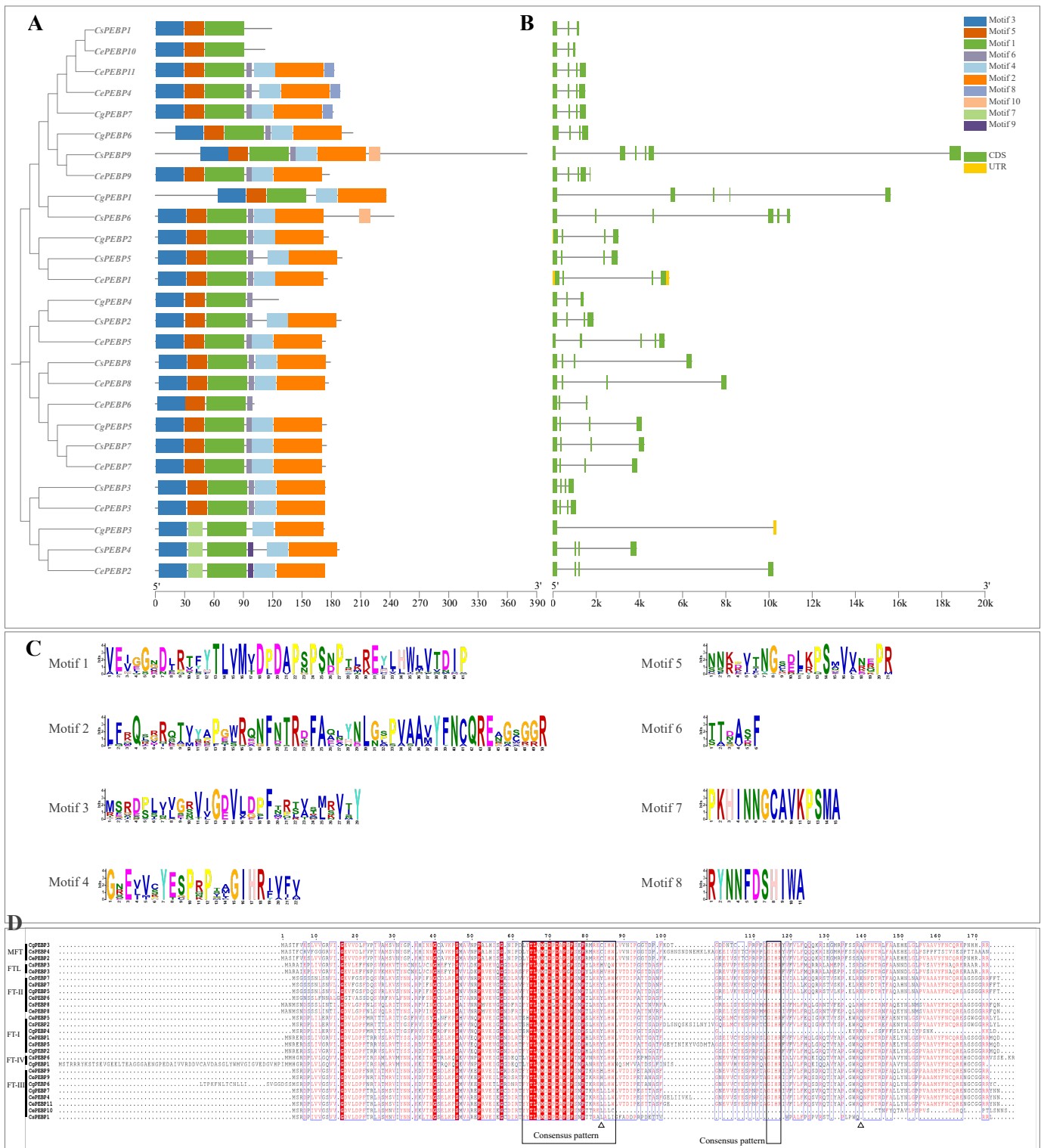

**Figure 3.** (**A**) The comparative map of *PEBPs* is based on the phylogenetic tree and conserved protein motifs of the three *Cymbidium* species. (**B**) Distribution of UTRs and CDSs of PEBP gene family members of the three *Cymbidium* species. Green represents CDS s and yellow represents UTRs. The scale at the bottom is used to compare the lengths of different genes and proteins. (**C**) Conserved domains of the three *Cymbidium* species protein sequences. The overall height of each stack indicates the sequence conservation at that position. (**D**) The *PEBP* homeodomain sequence alignment analysis of three *Cymbidium* species. The red blocks represent highly conserved residues. The red blocks represent highly conserved residues. Ce: *Cymbidium ensifolium*; Cg: *C. goeringii*; Cs: *C. sinense*.

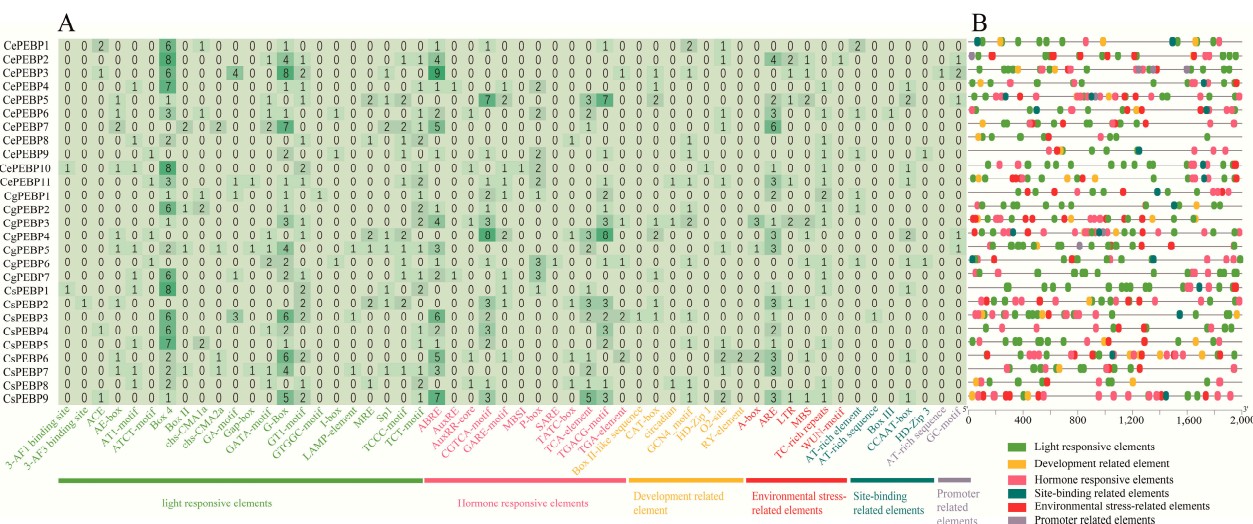

**Figure 4.** (**A**) Classification and statistics of cis-acting elements of three *Cymbidium* species. The numbers in the grid represent the number of elements, with darker colors indicating a larger number and lighter colors indicating a smaller number. (**B**) The promoter region's distribution of cis-acting elements of three *Cymbidium* species. The various types of cis-acting elements are represented by different colors and shapes. The sequence direction and length are indicated by the ruler at the bottom. Ce: *Cymbidium ensifolium*; Cg: *C. goeringii*; Cs: *C. sinense*.

### 3.6. Expression Patterns of PEBP Genes in C. ensifolium

Based on the transcriptome data, three *PEBP* genes were significantly expressed in the buds and flowers of *C. ensifolium*, and two genes were significantly expressed in the leaves and roots (Figure 5). Some genes (*CePEBP5*, *CePEBP1*) were expressed in several tissues, whereas some genes (*CePEBP8*, *CePEBP9*, and *CePEBP4*) showed little or no expression in the tissues. The average FPKM values of the transcriptome of *CePEBP* genes in different tissues of *C. ensifolium* are shown in Supplementary Table S2.

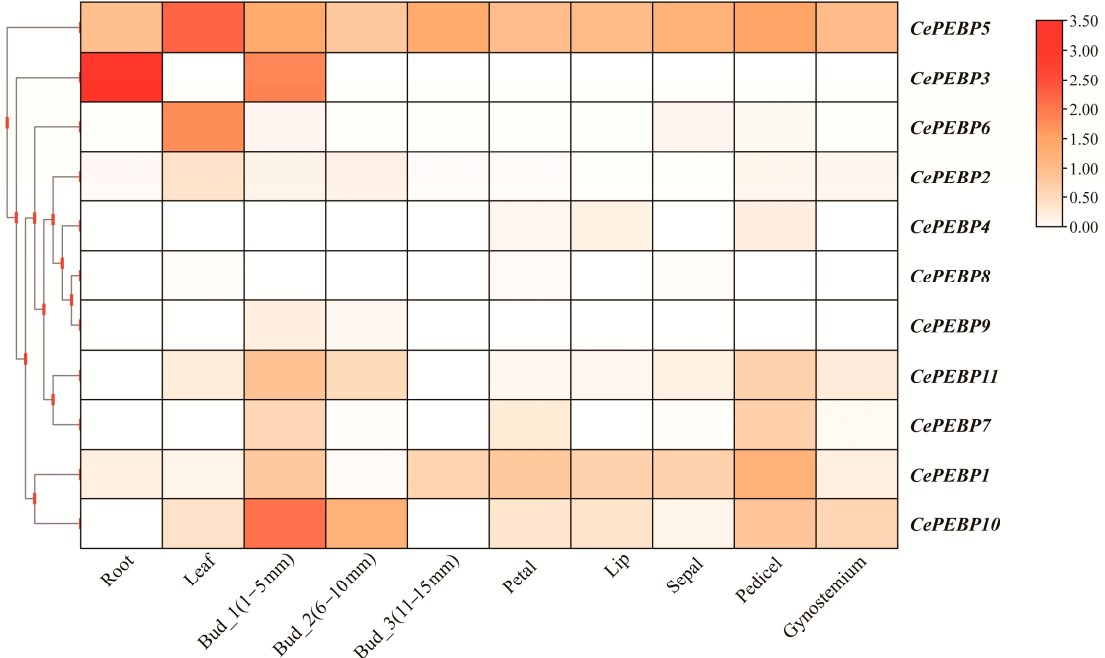

**Figure 5.** Expression of *CePEBPs* in various tissues of *Cymbidium ensifolium*. The dendrogram on the left displays the results of the inter-gene clustering analysis.

We selected four genes, *CePEBP1*, *CePEBP3*, *CePEBP5*, and *CePEBP10*, for RT-qPCR experiments according to the transcriptome data (primer sequence information is shown in Table S3). The RT-qPCR results showed that they were expressed highly in the bud (1–5 mm), leaves, roots, and pedicels, suggesting their vital role in multiple developmental stages of *C. ensifolium* (Figure 6). The transcriptome data and RT-qPCR results for *CePEBP1* and *CePEBP3* were basically the same. Transcriptome data indicated that *CePEBP5* was expressed in all tissues and had the highest expression in leaves. The RT-qPCR results indicated that it had the highest expression in the gynostemium. *CePEBP10* exhibited the highest expression in the bud (1–5 mm), while the RT-qPCR results showed the highest expression in leaves.

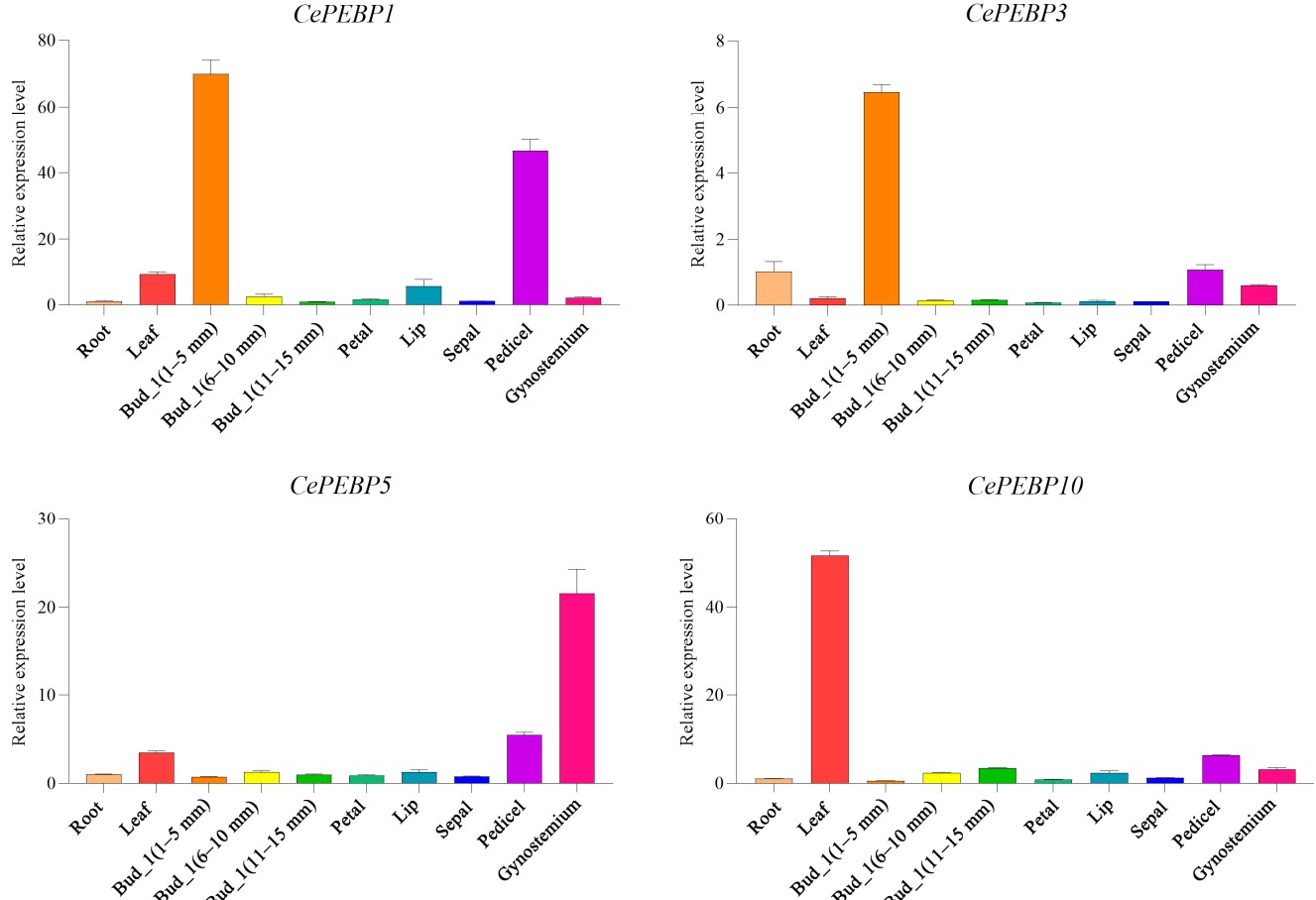

**Figure 6.** Analysis of gene expression of four *CePEBPs* in *Cymbidium ensifolium* at ten different organ materials.

## 4. Discussion

The growth habit, flowering time, flower number, and floral organ development of *Cymbidium* species are closely related to their ornamental value. Previous studies have cloned one PEBP homologous gene in each of the three *Cymbidium* species and found that *C. goeringii* might be primarily regulated by low temperatures, while *C. ensifolium* and *C. sinense* are regulated by the photoperiod [39]. This suggests that the study of the PEBP gene family may contribute to an improvement in ornamental traits in *Cymbidium* species. However, the role of the PEBP gene family in *Cymbidium* species has not been systematically studied to examine the common characteristics of its members.

In angiosperms, the PEBP gene family has undergone two ancient duplications, giving rise to three types: FT-like, TFL1-like, and MFT-like [4–6]. In this study, twenty-seven *PEBP* genes from three *Cymbidium* species were classified into three subfamilies (FT, TFL1, and

MFT) through phylogenetic analysis, which was consistent with other species [4–6]. The FT-like genes were the most diversified in terms of copy number among the three PEBP sub-families. In a study on the evolution of FT/TFL1 in tropical new orchids, the FT-like genes in monocotyledonous plants were divided into two branches: MonFT1 and MonFT2. Genes in the MonFT1 sub-branch might have played a role in delaying flowering, while genes in the MonFT2 sub-branch could have retained the function of promoting flowering [40]. Similarly to other monocots, three *Cymbidium* species carried more FT-like homologous sequences than TFL1-like and MFT-like, which could be further divided into four types (like I–IV). Among them, type I belongs to the MonFT2 sub-branch and promotes flowering through genes closely related to the *AtFT* gene, while types II, III, and IV belong to the MonFT1 sub-branch and inhibit flowering [40]. Additionally, previous studies have reported that the TFL1-like genes have undergone duplication during evolution in dicotyledons, followed by functional divergence from the TFL1 and CEN gene lineages [1,41]. In contrast to the FT genes, the TFL1 subfamily is more diverse in dicotyledons than in monocotyledons [40]. TFL1-like genes are either completely absent or very few are present in orchids, frequently as single copies. Only homology between *Oncidium* 'Gower Ramsey' and *Vanilla planifolia* has been reported [40,42,43]. This may be caused by the progressive loss of function of TFL1-like genes together with functional compensation by FT-like copies, but it still needs to be tested in the necessary experiments [40]. Among the three *Cymbidium* species, the absence of TFL1-like in *C. goeringii*, while *C. ensifolium* and *C. sinense* had a single copy, agrees with the results of previous studies [25,40]. These results reveal the functional differentiation and diversity in the PEBP gene family of three *Cymbidium* species.

PEBP protein has highly conserved D-P-D-x-P and G-x-H-R motifs in plants, and the binding of these motifs to anions is important for the conformation of the ligand-binding site of the PEBP protein [1,44]. Mutations close to this region may affect the binding of the PEBP protein with phosphate ions, thereby altering its interaction with FD [1,45]. Previous studies have indicated that a single amino acid determines the antagonistic activity of the floral regulators, including FT-like and TFL1-like. The residues Tyr85/His88 and Gln140/Asp144 in the FT-like and TFL1-like proteins may be the key residues that distinguish FT-like and TFL1-like activity, where they form hydrogen bonds in TFL1-like but not in FT-like [18,20]. For example, one amino acid substitution (replacing His-88 with Tyr in TFL1-like) can convert TFL1-like into FT-like, which promotes flowering [18]. In another study, specific mutations at the Glu-109, Trp-138, Gln-140, and Asn-152 sites can convert the FT-like into the TFL1-like, which inhibits flowering [19]. In this study, 27 *PEBPs* were identified in three *Cymbidium* species, and the results of the conserved motifs of all *PEBPs* indicated that these genes contained not only key amino acid residues but also two conserved motifs (D-P-D-X-P and G-X-H-R) [44,46]. Among them, the aa85 position of the FT-like III branch (*CgPEBP6*, *CsPEBP9*, *CePEBP9*) and five genes (*CePEBP4*, *CePEBP10*, *CePEBP11*, *CgPEBP7*, *CsPEBP1*) were replaced by His (H) and Leu (L) instead of Tyr (Y). Moreover, the aa140 (Gln, Q) of the FT-like II members was replaced by (Glu, E) and (His, H). FT-like II and III belong to the MonFT1 subbranch, which inhibits flowering in plants [40]. This indicates that the changes in Tyr85/His88 and Gln140/Asp144 residues of the PEBP gene family of three *Cymbidium* species can determine the functional conversion of FT/TFL1, which is similar to the results of previous researches [18,19,40].

Flowering is a key developmental process for environmental adaptation and reproduction in higher plants and requires a complex network of signaling pathways, which has been studied in many plants [1]. PEBP functions as a gene hub, integrating the photoperiodic pathway, vernalization pathway, autonomous pathway, gibberellin pathway, and age pathway in major floral induction pathways [1,47–49]. Investigating the transcriptional regulation of gene expression at the level of promoters by cis-acting elements has advanced our basic understanding of gene regulation and enriched the arsenal of readily available promoters [50]. In this study, a series of functional regulatory elements in the promoter region of the *PEBPs* were identified in three *Cymbidium* species, including growth and development factors, stress response factors, and plant hormone response factors. Among

them, the light-responsive elements had the maximum number, indicating that the *PEBPs* might be regulated by light signals and growth and development (Supplementary Table S1). Previous studies also found that *C. ensifolium* and *C. sinense* were regulated by photoperiod, while *C. goeringii* was regulated by other factors, such as low temperatures. The research results also indirectly confirmed this point [39].

The PEBP transcripts are abundant in numerous organs during the growth and development of orchids [25,42,51]. In *Phalaenopsis* 'Little Gem stripes', transcription of the *PEBP* genes among the various organs is detected. *PhFT1* is mainly expressed in vegetative buds, *PhFT2* is specifically expressed in leaves, and the expression level of *PhFT3* is highest in inflorescence [25]. But the FT homologous gene is highly expressed during flower organ development and growth processes in *Dendrobium* 'Chao Praya Smile' [51]. In *Oncidium* 'Gower Ramsey', the *OnFT* mRNA is widely detected in different organs at different growth stages and had the highest level in tender flower buds (2 mm) [42]. In this study, we observed that certain *CePEBP* genes exhibited tissue-specific expression (Figure 6), with *CePEBP3*, *CePEBP5*, and *CePEBP6* being specifically expressed in the developing roots and leaves of *C. ensifolium*, independently. *CePEBP1*, *CePEBP3*, *CePEBP5*, and *CePEBP10* were expressed specifically in buds and flowers, which might be related to flower differentiation and development. None of *CePEBP8*, *CePEBP9*, and *CePEBP4* were expressed in any of the tested tissues or organs, indicating that they were not expressed in *C. ensifolium*. The result of RT-qPCR analysis showed that *CePEBP1* and *CePEBP*3 had high expression in flower buds (1–5 mm) and pedicels; *CePEBP5* exhibited high expression in the gynostemium and pedicel; and *CePEBP10* showed high expression in the leaves of *C. ensifolium* 'Longyan Su'. This may be due to an incomplete correlation between sequencing and RT-qPCR samples. These findings also suggest that the various expression patterns of the PEBP gene family may contribute to further research on functional differentiation of the FT-like branch in orchids.

## 5. Conclusions

In this study, seven *CgPEBP*, nine *CsPEBP*, and eleven *CePEBP* were identified in three *Cymbidium* species, which were classified into three clades. The *PEBP* genes of *C. ensifolium* can play a significant role in the development and growth of the plant, particularly in the bud (1–5 mm). It was noteworthy that *CePEBP5* also played an indispensable role in both the vegetative and reproductive growth of *C. ensifolium*. The *CePEBP1* gene was crucial for root development, while *CePEBP1*, *CePEBP3*, *CePEBP5*, and *CePEBP10* might be involved in the growth and development of multiple floral organ tissues. These findings can provide possible directions for further investigations on the regulation of *PEBPs* on the flowering times of *C. ensifolium*.

**Supplementary Materials:** The following supporting information can be downloaded at: https://www.mdpi.com/article/10.3390/horticulturae10030252/s1. Table S1: Details of cis-acting elements in the promoter region of *PEBP* genes from three *Cymbidium* species (2000 bp upstream of the initiation codon); Table S2: Average FPKM value of *CePEBP* genes transcriptome in different tissues of *C. ensifolium*; Table S3: Primers used in this study.

**Author Contributions:** Conceptualization, methodology, J.C., F.W., Z.L. and D.P.; Software, formal analysis, visualization, writing—original draft preparation, J.C. and F.W.; Investigation, resources and data curation, validation, J.C., F.W., R.Z., Y.Z. and X.W.; Writing—review and editing, S.A., Y.A., Z.L. and D.P.; Supervision, J.C., F.W., Z.L. and D.P.; Project administration, D.P.; Funding acquisition D.P. All authors have read and agreed to the published version of the manuscript.

**Funding:** This research was funded by the National Natural Science Foundation of China (32071815).

**Data Availability Statement:** All data generated or analyzed during this study are included in this published article (Supplementary Files) and are also available from the corresponding author on reasonable request.

**Conflicts of Interest:** The authors declare no conflicts of interest.

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
