# Peer review of "Genome-Wide Identification of the PEBP Family Gene in Three Cymbidium Species and Its Expression Patterns Analysis in C. ensifolium"

_horticulturae, doi:10.3390/horticulturae10030252_

Round 1

Reviewer 1 Report

Comments and Suggestions for Authors

Dear authors,

The manuscript is well-written. To raise the quality of the manuscript, a few minor adjustments must be made. Kindly refer to the reviewer's report.

Best regards.

Author Response

Abstract

The abstract is skillfully composed and presented. The abstract's components are all expressed straightforwardly. However, I would suggest to rephrase the second sentence (Line 12-15).

Response: Thank you for your valuable comment. We have rephrased the second sentence on line 12-13.

Line 28: Cymbidium Species will be more appropriate

Response: Thanks a lot for your comments. We have revised it in the manuscript.

Introduction

Line 38: put space between “genes” and [4, 5].

Response: Thanks very much for your comments. We have reviewed these in the manuscript.

Line 41: delete full stop (periods) between seeds and [6-8].

Response: Thanks very much for your comments. We have reviewed these in the manuscript.

Line 41-43: Please rephrase “have been shown to be involved in the regulation” and put space before [9].

Response: Thanks very much for your comments. We have rephrased it in the manuscript.

Line 43: Erase full stop.

Response: Thanks very much for your comments. We have reviewed these in the manuscript.

Line 48-49: To render the sentence as clear as possible, please rephrase it.

Response: Thanks very much for your comments. We have rephrased it in the manuscript.

Line 51: Insert space between tissue and [12-16]. Also in line 56.

Response: Thanks very much for your comments. We have reviewed these in the manuscript.

Results

Line 72: please rephrase first sentence. (Eleven PEBP genes were found eleven in…)

Response: Thank you for your valuable comment. We have rephrased this sentence in the manuscript on line 146-148.

Line 77: Erase full stop. Also, you have to put Table 1 at the end of the sentence.

Response: Thank you for your valuable comment. We have revised it in the manuscript.

Line 84: if you use abbreviations, it would be nice to write them in the table as well, either below the table or right after the parameter.

Response: Thank you for your valuable comment. We have revised it in the table.

Line 87-88: It would be more appropriate to put this sentence at the top of this paragraph.

Response: Thank you for your valuable comment. We have revised it in the manuscript on line 145-146 base on the comment.

Line 125: Erase full stop.

Line 162: Please insert space after Arabidopsis.

Line 174 Erase space at the end of the sentence.

Line 195: replace “the” with “The”.

Response: Thank you for your valuable comment. We have revised them in the manuscript.

Line 224-225: please rephrase “The transcriptome data and the RT-qPCR results for CePEBP1 and CePEBP3 were basically the same.”

Response: Thank you very much for your professional comments. We have rephrased this sentence in the manuscript on line 298-299.

Line 230: erase full stop. Discussion

Line 236: Perhaps "seed germination" might be a better word choice than "germination."

Line 245, 252: insert space before brackets.

Line 291, 294: insert space before brackets. Please do this throughout the manuscript.

Response: Thank you for your valuable comment. We have revised them in the manuscript.

Material and methods

Line 328-333: it would be more appropriate to provide them (NCBI; SMART; ExPASy website) as a reference as well.

Response: Thanks very much for your comments. We have reviewed this sentence in the manuscript on line 85-91.

Line 340-383: provide full name of MEGA 7.0 software, version and the developer/manufacturer. Also, please do it for all software packages throughout the manuscript.

Response: Thanks very much for your comments. We have revised them in the manuscript.

Line 383: Please provide reference for the method.

Response: Thank you for your valuable comment. We have revised in the manuscript.

Conclusion

The conclusion is worded neatly and concisely.

Response: Thank you very much. We have revised the conclusion in the manuscript.

References

Please provide links for references.

Response: Thanks a lot for your valuable comment. We have revised this in the manuscript.

Supplementary files

Table 2: Please replace commas with full stops (periods) throughout the table.

Response: Thank you for your valuable comment. We have revised it in the Table 2.

Reviewer 2 Report

Comments and Suggestions for Authors

The article “Genome-wide identification of the PEBP family gene in three Cymbidium species and its expression patterns analysis in C. ensifolium”  by Chen et al is an interesting and well written manuscript. The PEBP gene family has been extensively studied in Arabidopsis thaliana and different plants.  All the family genes studied in different plants present very important functions in different developmental stages.  It is relevant to study this family in different species.  This study is a first approach of the characteristics of PEBP genes in three Cymbidium orchids and present data related the number PEBP genes and their position in the chromosome.  There are also data related to the function of this family that include conserved amino acid motifs, transcriptome analysis in different tissues and RT-qPCR analysis of gene expression in different tissues.

Minor corrections

 Please check all the scientific name, there are some that are not in cursives.  For example, the scientific name in lane 95, lane 258 and lane 259 are not in cursives.

You used either AA or aa. Please used only one form for all of them

Please give more information about the transcriptome analyses and include the number of replicates.

 Please describe the developmental stage of the buds of various sizes used in this study.

Author Response

Comment 1. Please check all the scientific name, there are some that are not in cursives. For example, the scientific name in lane 95, lane 258 and lane 259 are not in cursives.

Response: Thank you very much for your professional comments. We have checked carefully and revised them in the manuscript.

Comment 2. You used either AA or aa. Please used only one form for all of them

Response: Thank you very much for your professional comments. We have revised it in the manuscript.

Comment 3. Please give more information about the transcriptome analyses and include the number of replicates.

Response: Thank you very much for your professional comments. We have revised this in the manuscript on line 122-131.

Comment 4. Please describe the developmental stage of the buds of various sizes used in this study.

Response: Thank you very much. We have revised it in the manuscript (line 132).

Reviewer 3 Report

Comments and Suggestions for Authors

Dear Editor, the ms was well prepared and discussed, however a minor revision is required before the ms can be accepted for publication. All my comments are in the attached file.

Author Response

Comment 1. Line 67, “The results will help to further” -> “The objective of this work was to”.

Response: Thank you very much for your professional comments. We have rewritten it on line 64-66.

Comment 2. Line 70, The second section should Material and Methods, and not Results!!!

Response: Thank you very much for your professional comments. We have revised this point.

Comment 3. Line 89, It would be better if this gene names in this columns fit in only one line instead of two.

Response: Thank you very much for your professional comments. We have revised it.

Comment 4. Line 141, Line 219, Line 232, “C. goeringii”-> “Cymbidium goeringii”

Response: Thank you very much for your professional comments. We have revised it.

Comment 5. Line 176, The font of this Fig. 3. is so small that barely can be seen!

Response: Thank you for your valuable comment. In fact, Figure 3 was compressed in the manuscript. The original image of Figure 3 is clear and we will upload all the figures and Tables with a file namely ‘Figures and Tables’.

Comment 6. Line 204, The species names are missing in the title of this Fig. 4.

Response: Thank you for your valuable comment. We have added the species names in the title of this Figure 4.

Comment 7. Line 312, Provide full information about the location where the study was taken.

Response: Thank you very much for your professional comments. We have added the information on line 131-133.

Reviewer 4 Report

Comments and Suggestions for Authors

see PDF

Comments on the Quality of English Language

editing language required

Author Response

Line 4l, ...and dormancy of seeds [6-8].

Line 43, ...by FT-like genes.

Line 45, ... binding to Floring Locus D (FD) proteins.

Line 50, .. factor FLOWERING LOCUS D ( FD).

Lines 66-67...tissues, where expression of C, ensifolium occurred. The Introduction acceptably describes the functions of the PEBP family.

Line 72, ... found eleven in C. ensifolium.

Line 77, ...significantly.. The ...

Line 96, ... dispersed.. Ten...

Line 97, ... Chromosome 02,05, 09 (separate) contained ...

Lines 106, 109,113 ...(Figure. 1A).

Line 120,...chromosomes..(B)...

Line 163,...species (Figure. 3D). In ...

Line 178, ... is used to compare...

Line 184, ... three Cymbidium species...

Line 186, ... most cis-elements ...

Line 195, .. anaerobic induction (is not a phytohormone).

Line 195, ... respectively. The remaining...

Line 206, ... number.. (B) ...

Line 212, ....roots (Figure. 5).

Line 224,pedicels (Figure. 6).

Response: Thanks a lot for your professional comments. We have revised them in the manuscript.

Line 227,..the column (what means??).

Response: It means gynostemium, we have replaced “column” with “gynostemium” in the manuscript in order not to misunderstand the readers.

The Results section is 90% bioinformatics (see 4.1-4.6), which by no means diminishes the value of this work. However, my overall opinion of this draft will depend on the content and approach of the Discussion.

Response: Thank you very much for your professional comments. Orchids are also one of the most prestigious, horticulturally significant plants owing to their unique morphology and diversity. Despite PEBPs play an important role in regulating flowering, seed development and germination in plants, little is known about the features of PEBP genes in Cymbidium species. Thus, our results may provide a theoretical basis to uncover the underlying function of PEBP genes in three Cymbidium species and even other Orchidaceae.

Line 236, ..germination. [2, ... . The first paragraph is not significant in the overall context of the discussion. Must be deleted.

Response: Thanks a lot for your professional comments. We have deleted it in the manuscript.

Line 245, 252...flowering[23]. Square brackets should be separated from the word that precedes it.

Lines 287-88, ....Tyr85/His88 and 278 GInl40/Aspl44 residues (repeated several times in the text of this draft).

Lines 279-284, (repeated).

Lines 287-89,... repeated.

Line 300, ... of C. ensifolium(italic).

Response: Thanks a lot for your professional comments. We have revised them in the manuscript.

Line 378-379, repeated.

Response: Thanks a lot for your comments. We have revised them in the manuscript.

The quantitative real-time PCR (qRT-PCR) used (sec. 4.7.second paragraph)should be better described.

Response: Thank you very much for your suggestions. We have revised this paragraph in the manuscript.

The discussion contains only 5-6 new bibliographic citations in relation to the rest of the text, which significantly weakens this section. Advice to authors: consider reunifying the Results and Discussion sections. This approach can enhance the coherence and flow of the manuscript, facilitating a more seamless integration of findings and their interpretation, Conclusions: in line with the discussion, it is very weak.

Response: Thank you very much for your professional suggestions. We have revised the results and discussion sections in the manuscript.

Round 2

Reviewer 4 Report

Comments and Suggestions for Authors

I've nothing to add. 

Congratulations